# First Report on the Co-Occurrence and Clustering Profiles of Cardiovascular Lifestyle Risk Factors among Adults in Burkina Faso

**DOI:** 10.3390/ijerph19148225

**Published:** 2022-07-06

**Authors:** Kadari Cissé, Sékou Samadoulougou, Yves Coppieters, Bruno Bonnechère, Patrice Zabsonré, Fati Kirakoya-Samadoulougou, Seni Kouanda

**Affiliations:** 1Centre de Recherche en Epidémiologie, Biostatistique et Recherche Clinique, Ecole de Santé Publique, Université libre de Bruxelles (ULB), 1050 Brussels, Belgium; yves.coppieters@ulb.be (Y.C.); fati.kirakoya@ulb.be (F.K.-S.); 2Département Biomédical et Santé Publique, Institut de Recherche en Sciences de la Santé, Ouagadougou 03 BP 7192, Burkina Faso; senikouanda@gmail.com; 3Evaluation Platform on Obesity Prevention, Quebec Heart and Lung Institute Research Centre, Quebec, QC G1V 4G5, Canada; ouindpanga-sekou.samadoulougou.1@ulaval.ca; 4Centre for Research on Planning and Development (CRAD), Laval University, Quebec, QC G1V 0A6, Canada; 5REVAL Rehabilitation Research Center, Faculty of Rehabilitation Sciences, Hasselt University, 3590 Diepenbeek, Belgium; bruno.bonnechere@uhasselt.be; 6Unité de Formation et de Recherche en Sciences de la Santé, Université Joseph Ki-Zerbo Ouagadougou, Ouagadougou 03 BP 7021, Burkina Faso; zabsonre_pd@yahoo.fr; 7Institut Africain de Santé Publique, Ouagadougou 12 BP 199, Burkina Faso

**Keywords:** cardiovascular lifestyles risk factors, co-occurrence, clustering, Burkina Faso

## Abstract

The co-occurrence of cardiovascular risk factors is usually associated with a higher risk of cardiovascular disease (CVD) or cancer. This study aimed to determine the prevalence of the co-occurrence and its determinants and to identify the clustering profiles of lifestyle risk factors among the adult population in Burkina Faso. Among 4692 participants, 4377 adults from the first STEPS survey conducted in Burkina Faso were considered in this analysis. Four lifestyle risk factors (smoking, alcohol consumption, inadequate fruit and vegetable intake and low physical activity) were analysed. The clustering was evaluated using the observed/expected (O/E) ratio approach. To identify the determinants of co-occurrence, we performed a modified Poisson regression. The prevalence of the co-occurrence of two or more cardiovascular lifestyle risk factors was 46.4% (95% CI: 43.1–49.7). The main determinants of the co-occurrence were being male (adjusted prevalence ratio (aPR): 1.27 (95% CI: 1.16–1.38)), advanced age (55–64 years old: aPR: 1.45 (95% CI: 1.31–1.60)) and a high level of education (aPR: 1.29 (95% CI: 1.09–1.52)). The clustering profile for lifestyle risk factors was tobacco consumption combined with alcohol consumption (O/E: 2.77 (95% CI: 2.12–3.56)), and concurrent involvement in all four lifestyle risk factors (O/E = 1.51 (95% CI: 1.19–1.89)). This first population-based report on the co-occurrence of lifestyle risk factors calls for action to tailor health-promoting interventions to increase healthy lifestyle behaviors. The identified CVD-risk clustering should be considered as an important step in this strategy development in Burkina Faso.

## 1. Introduction

Cardiovascular disease (CVD) represents one of the most challenging public health issues worldwide. In 2016, the global estimated deaths caused by CVD were 17.9 million. More than three-quarters of these deaths occurred in low- and middle-income countries. Much of the burden of CVD is preventable through the modification of a set of cardiovascular risk factors that are classified as lifestyle risk factors (tobacco use, harmful alcohol consumption, low fruit and vegetable intake and physical inactivity) and metabolic risk factors (high blood pressure, diabetes, overweight/obesity and elevated total cholesterol) [1,2]. A recent study showed that cardiovascular lifestyle risk factors are the largest contributor to death and their impacts seem to be higher in low- and middle-income countries than in high-income countries [3]. 

These lifestyle risk factors can co-occur in the same individual and the co-occurrence of two or more risk factors is known to be associated with a greater risk of cardiovascular diseases or cancer than single risk factors [4,5]. The World Health Organisation (WHO) has prioritised these lifestyle risk factors in their global action plan to prevent non-communicable diseases (NCD) [6]. To achieve this global goal, it is useful to understand whether the risk factors are linked, how these risk factors are linked to each other and the determinants of this link. This may help to identify which determinants are linked and need to be targeted in simultaneous lifestyle risk factors interventions to successfully decrease the risk. These interventions may have beneficial effects on many NCD since these risk factors are drivers of the current trend in NCD in low- and middle-income countries [7]. Interventions addressing lifestyle risk factors are known to be cost-effective [8,9], even if some doubts about their efficacy persist [10]. To plan successful interventions, it is important for public health policy makers to know which sociodemographic groups of the population are at high risk of the co-occurrence or clustering of lifestyle risk factors [11,12].

Previous studies have found a prevalence of the co-occurrence of two or more lifestyle risk factors, these being 52% in the USA, 68% in England, 55% in the Netherlands and 59% in Brazil [11]. In Sub-Saharan Africa (SSA), studies on the co-occurrence and clustering of risk factors are rare. A study in Kenya reported that one-fifth (19.8%) of adults in urban areas had the co-occurrence of lifestyle risk factors [13].

This study sought to examine the co-occurrence and clustering of cardiovascular lifestyle risk factors among the adult population in Burkina Faso, a West Africa country, to help to develop targeted approaches and interventions to mitigate the effects of these risk factors. Regarding the levels of prevalence of the metabolic (intermediate) risk factors and the high burden of NCD, we hypothesized that there is a high level of co-occurrence of these risk factors and that this co-occurrence highly contributes to rise the morbidity and mortality due to NCD in the country. 

## 2. Materials and Methods

### 2.1. Study Setting, Type, and Population

We performed an analysis of the data from the first nationwide population-based STEPS survey conducted in Burkina Faso in 2013 (before the implementation of the first NCD plan drafted in 2016). Burkina Faso is located in SSA. In 2019, the population was 20,870,060 inhabitants and the life expectancy at birth was 61.6 years. The urbanity rate increased from 12.7% in 1985 to 22.7% in 2006 and it reached 26.3% in 2019. The country is facing an increasing burden of NCD, including CVD, in addition to the existing high burden of infectious diseases. This change in the disease pattern motived the country to draft an NCD program in 2016. Before implementing this programme, the STEPS survey was conducted in 2013 to provide a baseline picture of the NCD risk factors. Data were drawn from the World Health Organization’s Stepwise approach to NCD risk factor surveillance (STEPS). The STEPS survey was a cross-sectional survey, that was designed to obtain a regional representative sample of the general population aged 25–64 years. Briefly, a stratified sampling method was adopted to reflect the representation of urban and rural areas in the 13 administrative regions of the country. In each stratum, a three-stage cluster sampling was performed. In the first stage, enumeration areas (EAs) were selected through random sampling with a probability proportional to the size of the EAs per administrative region. In the second stage, a uniform sample of 20 households from the listed households in each cluster was selected using a simple systematic sampling. In the third stage, an individual aged 25–64 years was randomly selected in each household using the method of Kish. A total of 240 EAs were selected (186 EAs for rural areas and 54 EAs for urban areas). Of the 4800 individuals selected, 4691 participated in the interview. In total, 4377 had complete information on the CVD lifestyle risk factors and were finally included in the present analyses. 

### 2.2. Data Collection and Measurement

Data were collected between 26 September and 18 November 2013 by comprehensively trained health workers using the standard WHO STEPS instrument. Information on the socio-demographic conditions and lifestyle risk factors (tobacco use, alcohol use, fruit and vegetable intake and physical activity level) was used in this study and was obtained using related questionnaires. All the lifestyle variables were measured using standard methods as described in the WHO STEPS manual [14]. 

### 2.3. Study Variables

The study variables were grouped into sociodemographic and lifestyle risk factors variables.

#### 2.3.1. Sociodemographic Variables

Sociodemographic variables, namely, sex (women/men), age (grouped as a categorical variable: 25–34, 35–44, 45–54 and 55–64 years) and marital status (single/married), were considered for this analysis. Furthermore, we considered geographic variables (residence (urban/rural) and health region) and socioeconomic proxies (education (did not attend school; primary, secondary and more) and profession (wage earner; self-employed and jobless).

#### 2.3.2. Lifestyle Risk Factors

The first step of the STEPS survey involved asking questions about demographic information and lifestyle risk factors, specifically smoking, alcohol consumption, fat consumption, fruit and vegetable intake and physical activity. In this analysis, we defined as current tobacco users, participants who reported that they are consuming tobacco products (smoking and smokeless). Current alcohol users were defined as those who consumed alcohol within 30 days before the survey. Inadequate fruit and/or vegetable intake was considered if the participants consumed fewer than five servings (400 g) per day according to the WHO definition (the intake of five servings of fruits and vegetables each day is associated with a lower risk of dying from cardiovascular disease, cancer and respiratory disease [15,16]). Physical activity was measured using the metabolic equivalents of task (METs) minutes per week. A low level of physical activity (physically inactive) was defined according to the WHO guidelines [17].

### 2.4. Statistical Analysis

Four lifestyle risk factors were analysed in this study: current tobacco use, current alcohol use, inadequate fruit and/or vegetable intake and low physical activity. All the variables were categorized as binary variables, with 0 indicating the “absence of risk” and 1 indicating the “presence of risk”. To analyse multiple risk factors, two approaches are usually used by researchers: the co-occurrence of two or more risk factors and the clustering of risk factors [12]. To identify the cluster profiles of the lifestyle risk factors, the observed/expected ratio method was chosen since it is simpler to interpret in a clinical setting than factor analysis [12]. The ratio was obtained by dividing the observed prevalence of each profile of risk factors by the expected prevalence. The expected prevalence of each profile was computed assuming that the risk factors occurred independently. Thereby, the probability of each combination was obtained by multiplying the probabilities of each risk factor included in the profile (the weighted prevalence of each risk factor in the study population was used as the probabilities), based on its distribution in the study population. The risk factors in each combination were considered significantly clustered if the ratio was greater than 1 (i.e., the confidence interval of the ratio did not include 1). The statistical significance of the clustering was assessed by computing the confidence interval of the ratio using Newton’s method assuming a Poisson distribution. A modified Poisson regression, obtained through a generalised estimating equation, was used to derive the prevalence ratio, in order to investigate the association between the sociodemographic variables and the co-occurrence of two or more risk factors. All the analyses were performed using Stata 15.1.

## 3. Results

### 3.1. Characteristics of the Study Population 

Overall, 4377 individuals were included in the analysis, of which 54.6% were women, 42.2% were aged 25–34 years, 87.6% were married, 72% were rural residents and 77.6% had no formal education (Table 1).

### 3.2. Prevalence and Determinants of the Co-Occurrence of Lifestyle Risk Factors

Among the study population, 19.3% (95% CI: 17.2–21.6) consumed tobacco (smoking and smokeless tobacco), 27.8% (95% CI: 24.6–31.3) were current alcohol consumers, 95.8% (95% CI: 94.1–97.0) had inadequate fruit and vegetable intake and 14.9% (95% CI: 12.6–17.6) were physically inactive. As shown in Table 2, the prevalence of the co-occurrence of two or more risk factors was 46.4% (95% CI: 43.1–49.7). The prevalence of two, three and four lifestyle risk factors were 34.2%, 10.8%, and 1.4%, respectively. The co-occurrence of alcohol consumption and inadequate fruit and vegetable intake was most prevalent (observed prevalence was 15.4%) followed by tobacco consumption and inadequate fruit and vegetable intake (observed prevalence was 9.7%). The co-occurrence of inadequate fruit and vegetable intake and low physical activity was 8.62%.

As presented in Table 3, the main sociodemographic groups with higher prevalence of co-occurrence of lifestyle risk factors were male, advanced age and higher level of education. Thus, adjusted for all the potential confounders (age group, level of education, marital status, profession and residence), we found that the prevalence of the co-occurrence of lifestyle risk factors was 1.27 times higher(adjusted prevalence ratio (aPR) = 1.27 (95% CI: 1.16–1.38) among men compared to women. Individuals who were 55–64 years old had a 1.45 times higher prevalence of co-occurrence compared to those who were 25–34 years old (aPR: 1.45 (95% CI: 1.31–1.60)). The prevalence was higher among individuals who attended at least secondary school (aPR: 1.29 (95% CI: 1.09–1.52)) compared to those who were unschooled. After adjusting for all the variables in Table 1, we did not find any difference between urban and rural areas regarding the co-occurrence of lifestyle risk factors that was significant at the 5% level.

### 3.3. Clustering Profiles of the CVD Risk Factors

Regarding the clustering profile of the lifestyle risk factors, we found three significant lifestyle profiles. The first profile combined tobacco use and alcohol consumption (O/E = 2.77 (95% CI: 2.12–3.56)); the second combined tobacco use, alcohol consumption and inadequate fruit and vegetable intake (O/E = 1.58 (95% CI: 1.44–1.73)); and the last one combined all of the four lifestyle risk factors (tobacco use, alcohol consumption, inadequate fruit/vegetable intake and physical inactivity (O/E = 1.51 (95% CI: 1.19–1.89)) (Table 4).

## 4. Discussion

### 4.1. Key findings

Despite the availability of a database of the first nationwide survey on the common risk factors for non-communicable disease, including cardiovascular disease, this is, to the best of our knowledge, the first study on the co-occurrence and clustering of cardiovascular risk factors in Burkina Faso. We reported a high prevalence of co-occurrence of risk factors. The co-occurrence of alcohol consumption and inadequate fruit and vegetable intake on one hand and tobacco consumption and inadequate fruit and vegetable intake on other hand were mostly prevalent. We also found significant cluster profiles of the lifestyle risk factors, suggesting that some risk factors are linked and occur most dependently. We found three main clusters: (i) tobacco use + alcohol consumption; (ii) tobacco use + alcohol consumption + inadequate fruit and vegetable intake; and (iii) tobacco use + alcohol consumption+ inadequate fruit and vegetable intake + physical inactivity.

### 4.2. Co-Occurrence of Lifestyle Risk Factors 

Our study reports that nearly half (46.4%) of the adult population is engaged in at least two cardiovascular lifestyle risk factors. The number of studies on the co-occurrence of multiple lifestyle risk factors and their clustering has increased over the last years, highlighting the importance of studying these aspects [12]. Most recent papers have reported co-occurrence as a stage prior to the analysis of the clustering of risk factors [12], as we have done in this study. However, different approaches are used to analyse the co-occurrence and clustering, and different combinations of risk factors (often lifestyle and metabolic risk factors) are selected depending on the authors. These situations complicate comparisons between studies [12]. 

Different prevalence of co-occurrence of the risk factors have been reported in SSA, depending on the number of risk factors considered and the study area or population. Thus, in Ethiopia, Zenu et al. [18] reported in 2020, that 65.5% of adults had two or more of the four lifestyle risk factors. Haregu et al. noted that 19.8% of their study participants had the co-occurrence of lifestyle risk factors in Nairobi (Kenya) [13]. In Uganda, Wesonga et al. reported that 56.4% of adults had two or more risk factors [19]. Wesonga et al.’s study considered a high body mass index and raised blood pressure in addition to the four lifestyle risk factors. The prevalence of the co-occurrence was close to 38.5% among the adult population in Brazil [20]. Looking at the literature in developed countries, it can be seen that, in the UK, the prevalence of the co-occurrence of the four lifestyle risk factors was close to 36.9% in 2016, as reported by Birch et al. [21]. Globally, the co-occurrence seems to be a major public health issue in all parts of the world. This situation is worrying since the adolescent population is affected [22]. Indeed, Uddin et al. [22] had analysed data from 89 countries and found out that 34.9% of adolescents had at least three of the following risk factors: current smoking, current alcohol drinking, low fruit and vegetable intake, physical inactivity, overweight/obesity and high sedentary behaviour. This rate seems to be high in different WHO regions (23.3% in Africa and 56.2% in the Americas), suggesting that the co-occurrence of lifestyle risk factors is a universal phenomenon regardless of age groups [22]. In our study, the current level of the co-occurrence of risk factors could be attributed to the high prevalence of individual risk factors. Indeed, a previous study has reported a high prevalence of tobacco consumption [23]. Inadequate fruit and vegetable intake was the most prevalent lifestyle risk factor among the adult population in low-income countries. Indeed, in Kenya [24], Bangladesh [25] and other low-income countries [26], at least 70% of the adult population had insufficient fruit and vegetable intake. We found the same result. Low physical activity is known to be less prevalent in the low-income context [27]. In this study, 14.9% of adults were physically inactive. Globally, the prevalence of physical inactivity is close to 27.5%, with an overall prevalence of 16.2% in low-income countries [28]. Approximately 27.8% of the adult population in Burkina Faso are current alcohol drinkers. This percentage approaches the overall prevalence of global alcohol consumption, which was close to 32.5% in 2016 [29]. The current levels of lifestyle risk factors and their co-occurrence signal a future epidemic of the related diseases (cardiovascular disease and cancer). To curb the rising burden of cardiovascular disease, it is important to promote lifestyle-change interventions, which may be cost-effective if they address multiple behaviours simultaneously. To better guide further strategies, we have performed a cluster analysis, since the co-occurrence analyses do not provide information about how the lifestyle risk factors are linked [12]. 

Our study reported that the main sociodemographic groups with higher risk of the co-occurrence of lifestyle risk factors for cardiovascular disease are males, older individuals and an increased level of education. The prevalence of the co-occurrence of lifestyle risk factors seems to increase with age in all regions [22,30]. Consistent with other studies [24,25], we found that men had a higher prevalence of the co-occurrence of two or more lifestyle risk factors. However, many other studies have noted that women have a higher prevalence [13,18,30]. It is known that men are more likely to engage in risky lifestyle choices, such as smoking or alcohol consumption. Consistent with previous study findings [30], in this study, we did not find a significant difference in the prevalence of the co-occurrence of two or more risk factors between rural and urban areas. This finding might be explained by the fact that, in rural areas, many women engage in smokeless tobacco use (which was included in this study to define tobacco used) [23,31]. In addition, low fruit and vegetable consumption and alcohol consumption are also prevalent in rural areas [32]. The association between education level and cardiovascular risk factors has been discussed in the scientific literature. Rosengren et al. [33] noted that people with low levels of education in low- and middle-income countries had lower risk factor levels but a higher risk of cardiovascular disease compared to those with higher levels of education. They called this observation an “apparent paradox” [33]. 

### 4.3. Clustering Profiles of Lifestyle Risk Factors

Our study found that tobacco use + alcohol consumption, tobacco use + alcohol consumption + inadequate fruit and vegetable intake, and tobacco use + alcohol consumption+ inadequate fruit and vegetable intake + physical inactivity were the three main clusters of lifestyle risk factors identified among the adult population in Burkina Faso. The clustering of the lifestyle risk factor has been commonly reported in the literature [34]. A systematic review conducted by Meader et al. [11] in 2016 showed that alcohol misuse and smoking is the most common risk behaviour cluster among the adult population. This systematic review also shows that the clustering between smoking and alcohol use and smoking and low fruit and vegetable intake (“unhealth diet”) was the most prevalent among the adult population [11]. A similar result was reported in another context [35]. Adults who consumed tobacco products or alcohol were more likely to be sedentary or have a poor diet [11]. The same findings were observed in study targeting adolescents [22]. Thus, the clusters identified in our study seem to be a “universal phenomenon” in many countries [22]. We are aware that some clusters, such as tobacco use + alcohol consumption, are already known. However, this study highlighted new clusters, such as tobacco use + alcohol consumption + inadequate fruit and vegetable intake and tobacco use + alcohol consumption+ inadequate fruit and vegetable intake + physical inactivity. These new clusters need to be targeted in lifestyle interventions. These interventions should not be limited to only increasing the knowledge of the population regarding cardiovascular risk factors, but they have to be integrated into risk assessments. As Burger et al. [36] have shown, good CVD knowledge may not be sufficient for people to change their own cardiovascular risk factor levels. They have to be aware of their own risk. 

### 4.4. Strength and Limitations

This study has some limitations that need to be considered when interpreting the results. In this study, we dichotomized all lifestyle variables and the number of lifestyle factors. This may have caused a loss of precision. However, it has the advantage of facilitating the interpretation of the results. Furthermore, there are different approaches to analysis clustering, which complicate comparisons between studies. The lifestyle variables were self-reported, which may have caused social desirability bias and recall bias. However, the data collectors were well trained to avoid these biases. We are aware that 8 years have elapsed since the data collection. However, this study provides, for the first time, the pattern of the co-occurrence and clustering of cardiovascular lifestyle risk factors before the NCD plan started in 2016 in Burkina Faso. The second survey, which is ongoing, might represent an opportunity to evaluate the changes in the patterns of lifestyle risk factors after the initiation of the NCD plan.

## 5. Conclusions

This first population-based study reported a high prevalence of the co-occurrence of cardiovascular lifestyle risk factors particularly among men, elders and those with high levels of education in Burkina Faso. The results show that people who engaged in one lifestyle risk factor also seem to have an increased risk of engaging in others risk factors. Such patterns reinforce the need for health-promoting interventions to take a holistic approach toward targeting multiple behaviours. These efforts might specially target men, older people and those with high levels of education in Burkina Faso. 

## Figures and Tables

**Table 1 ijerph-19-08225-t001:** Sociodemographic profile of the study population.

Characteristics	All *n* (% *)	Women *n* (% *)	Men *n* (% *)
All participants	4377 (100.0)	2287 (100.0)	2090 (100.0)
Sexes			
Women	2287 (54.6)		
Men	2090 (45.4)		
Age group, years			
25–34	1996 (42.2)	1110 (45.5)	886 (38.1)
35–44	1092 (27.7)	568 (27.4)	524 (28.1)
45–54	783 (18.7)	390 (17.4)	393 (20.2)
55–64	506 (11.5)	219 (9.7)	287 (13.6)
Marital status			
Single	593 (12.4)	270 (11.1)	323 (14.0)
Married	3784 (87.6)	2017 (88.9)	1767 (86.0)
Completed level of education			
No formal school	3384 (77.6)	1853 (81.3)	1531 (73.1)
Primary school	685 (15.4)	298 (13.1)	387 (18.2)
Secondary or higher	308 (7.0)	136 55.6)	172 (8.7)
Profession			
Wage earner	244 (5.6)	77 (3.3)	167 (8.4)
Self-employed	3006 (65.9)	1180 (49.2)	1826 (86.1)
Jobless	1127 (28.4)	1030 (47.5)	97 (5.5)
Residence			
Urban	990 (28.0)	553 (29.6)	437 (26.0)
Rural	3387 (72.0)	1734 (70.4)	1653 (74.0)

Note: *, weighted percentage.

**Table 2 ijerph-19-08225-t002:** Prevalence of the co-occurrence of lifestyle risk factors by sex.

Characteristics	*n*	Overall	Men	Women
		% (95% CI)	% (95% CI)	% (95% CI)
All participants	4377	46.4 (43.1–49.7)	53.7 (49.8–57.6)	40.3 (36.5–44.2)
Sex				
Women	2287	40.3 (36.5–44.2)		
Men	2090	53.7 (49.8–57.6)		
Age group, years				
25–34	1996	41.5 (37.5–45.7)	53.6 (48.5–58.7)	33.1 (28.2–38.3)
35–44	1092	44.6 (40.1–49.1)	50.2 (44.3–56.1)	39.7 (34.4–45.4)
45–54	783	52.6 (47.3–57.8)	53.5 (46.5–60.4)	51.7 (44.1–59.2)
55–64	506	58.5 (52.6–64.1)	61.4 (53.8–68.5)	55.0 (44.1–59.2)
Marital status				
Single	593	54.7 (48.3–61.0)	61.0 (52.9–68.5)	48.1 (39.5–56.8)
Married	3784	45.2 (41.8–48.6)	52.5 (48.4–56.6)	39.3 (35.3–43.4)
Completed level of education				
No formal school	3384	44.0 (40.4–47.7)	50.6 (46.2–55.0)	39.1 (34.8–43.5)
Primary school	685	51.4 (46.2–56.5)	58.6 (51.4–65.5)	43.0 (36.6–49.7)
Secondary or higher	308	61.7 (52.8–70.0)	69.6 (59.1–78.4)	51.5 (40.7–62.0)
Profession				
Wage earner	244	58.3 (50.0–66.2)	64.5 (55.5–72.6)	45.3 (31.2–60.2)
Self-employed	3006	48.6 (44.5–52.7)	52.3 (47.9–56.6)	43.2 (37.3–49.2)
Jobless	1127	38.9 (34.5–43.5)	59.4 (45.3–72.0)	36.9 (32.4–41.7)
Residence				
Urban	990	51.6 (45.8–57.3)	62.1 (55.5–68.2)	43.9 (37.1–50.9)
Rural	3387	44.3 (40.4–48.3)	50.7 (46.1–55.4)	38.8 (34.2–43.6)

**Table 3 ijerph-19-08225-t003:** Factors associated with the co-occurrence of lifestyle risk factors using a modified Poisson regression with generalized estimating equations.

Characteristics	*n*	PR (95% CI)	*p*	aPR (95% CI)	*p*
All participants	4377				
Sex			<0.001		<0.001
Women	2287	1		1	
Men	2090	1.36 (1.25–1.48)		1.27 (1.16–1.38)	
Age group, years			<0.001		<0.001
25–34	1996	1		1	
35–44	1092	1.11 (0.99–1.22)		1.10 (1.01–1.20)	
45–54	783	1.23 (1.10–1.38)		1.23 (1.12–1.35)	
55–64	506	1.48 (1.31–1.68)		1.45 (1.31–1.60)	
Marital status			0.024		0.054
Single	593	1		1	
arried	3784	0.87 (0.78–0.98)		0.90 (0.82–1.00)	
Completed level of education			0.026		0.014
No formal school	3384	1		1	
Primary school	685	1.04 (0.92–1.17)		1.05 (0.95–1.16)	
Secondary or higher	308	1.25 (1.06–1.48)		1.29 (1.09–1.52)	
Profession			<0.001		0.10
Wage earner	244	1		1	
Self-employed	3006	0.93 (0.78–1.12)		1.09 (0.93–1.26)	
Jobless	1127	0.72 (0.58–0.87)		0.97 (0.81–1.14)	
Residence			0.147		0.071
Urban	990	1		1	
Rural	3387	0.86 (0.71–1.05)		0.90 (0.80–1.01)	

Note: PR = prevalence ratio; aPR = adjusted prevalence ratio.

**Table 4 ijerph-19-08225-t004:** Clustering of lifestyle risk factors.

Number of Risk Factors	Tobacco Use	Alcohol Use	Inadequate Fruit and Vegetable Intake	Low Physical Activity	All (*n* = 4377)			Women (*n* = 2287)			Men (*n* = 2090)		
					O (%)	E (%)	O/E (95% CI)	O (%)	E (%)	O/E (95% CI)	O (%)	E (%)	O/E (95% CI)
0	−	−	−	−	2.47	1.98	1.24 (1.04–1.47)	2.89	2.41	1.20 (0.95–1.49)	2.01	1.64	1.24 (0.92–1.62)
1	+	-	−	−	0.39	0.47	0.82 (0.45–1.38)	0.22	0.32	0.69 (0.12–1.75)	0.57	0.66	0.92 (0.46–1.64)
1	−	+	−	−	1.03	0.76	1.36 (1.04–1.74)	1.49	0.79	1.89 (1.44–2.0)	0.53	0.76	0.70 (0.31–1.43)
1	−	−	+	−	49.47	47.58	1.04 (1.00–1.08)	55.53	52.63	1.06 (0.99–1.11)	42.84	40.00	1.07 (1.00–1.14)
1	−	−	−	+	0.27	0.35	0.77 (0.34–1.42)	0.48	0.51	0.94 (0.43–1.67)	0.05	0.28	0.0 (0.0–3.69)
2	−	−	+	+	8.62.	8.34	1.03 (0.93–1.14)	10.28	11.17	0.92 (0.80–1.05)	6.80.	6.83	0.99 (0.83–1.17)
2	+	+	−	−	0.50	0.18	2.77 (2.12–3.56)	0.26	0.10	2.55 (1.40–4.12)	0.77	0.30	2.57 (1.79–3.40)
2	+	−	−	+	0.02	0.08	0.25 (0.0–3.69)	0.0	0.0		0.05	0.11	0.45 (0.00–3.69)
2	+	−	+	−	9.69	11.38	0.85 (0.76–0.94)	4.63	6.9	0.67 (0.52–0.84)	15.22	16.03	0.95 (0.85–1.06)
2	−	+	−	+	0.09	0.13	0.68 (0.15–2.19)	0.09	0.17	0.52 (0.01–2.78)	0.10	0.13	0.77 (0.01–2.79)
2	−	+	+	−	15.43	18.32	0.84 (0.77–0.91)	15.78	17.11	0.92 (0.83–1.03)	15.03	18.58	0.81 (0.71–0.91)
3	−	+	+	+	2.65	3.21	0.84 (0.67–1.01)	2.32	3.63	0.64 (0.44–0.89)	3.02	3.17	0.95 (0.72–1.22)
3	+	−	+	+	1.28	2.00	0.64 (0.45–0.89)	1.31	1.46	0.90 (0.59–1.30)	1.24	2.73	0.46 (0.24–0.80)
3	+	+	+	−	6.92	4.38	1.58 (1.44–1.73)	3.76	2.23	1.68 (1.41–1.97)	10.39	7.44	1.39 (1.24–1.56)
3	+	+	−	+									
4	+	+	+	+	1.17	0.77	1.51 (1.19–1.89)	0.96	0.48	2.02 (1.49–2.74)	1.39	1.27	1.10 (0.75–1.56)

Note: O: observed prevalence; E: expected prevalence based on the assumption that the risk factors are independent. The clusters were considered statistically significant if the confidence interval of the ratio did not include 1. The background color and + mean that the lifestyle risk factor it present and − mean it is absent.

## Data Availability

The dataset of the STEPS survey that was used in this research is available at the Ministry of Health upon request to Bicaba Brice: bicaba_brico@yahoo.fr or Zoma Torez: torezo2000@yahoo.fr). All survey materials are available on the WHO website (https://extranet.who.int/ncdsmicrodata/index.php/catalog, accessed on 25 January 2021).

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
