# Peer review of "First Report on the Co-Occurrence and Clustering Profiles of Cardiovascular Lifestyle Risk Factors among Adults in Burkina Faso"

_ijerph, 2022, doi:10.3390/ijerph19148225_

Round 1

Reviewer 1 Report

The manuscript, entitled "First Report on the Co-occurrence and Clustering Profiles of Cardiovascular lifestyle risk factors among adults in Burkina Faso," shows the prevalence of the co-occurrence of several cardiovascular disease risk factors among the adult population in Burkina Faso. The authors make no hypotheses. The study aims to assess the prevalence of co-occurrence of the lifestyle risk factors of cardiovascular disease in the Burkina Faso adult population. The authors did not clearly explain what exactly they found. In the Discussion section, the co-occurrence of risk factors for cardiovascular pathologies is considered superficially, i.e., parallels between the study's data and other research data are not fully drawn. The conclusions of the gender comparison are not supported by statistical data and cannot be considered reliable.

Reviewer 2 Report

Dear authors,

Thank you for the opportunity to review your manuscript. The manuscript is well written, coherent and easy to read. Below are some observations and comments that I hope you find useful to strengthen your paper.

  1. My main overall feeling after reading your manuscript is, ‘so what?’. As you have indicated, the results you have found are replicated in many other studies. We have known for a long time know that alcohol and smoking are often concurrent behaviours. We also know that taking a multifactorial approach to risk reduction is more efficient and important than a single factor approach. It is not clear to me how this manuscript adds substantial novel value, apart from the location of the study. How will your study help to improve risk factor control, policy making, program development, etc.?
  2. The data comes from the 2013 STEPS survey – this data is almost a decade old now. How relevant is the data for contemporary public health policy making?
  3. It is not clear what ‘wage earner’ means. Do you mean an employee of a business? Does the jobless group include those who cannot work (e.g. disability) and those who choose not to work (e.g. retired).? If so, does this impact on the outcomes?
  4. It’s not clear to me what the adjusted prevalence ratio is? I am assuming you are adjusting for the potential confounders: age, sex, marital status, education, profession and residence but the text between lines 167-176 suggests there were two sets of adjustment. Please clarify.
  5. Why have you excluded current non-daily smokers? I understand that you have chosen to use a dichotomous outcome (which is fine), but why not combine daily and non-daily smokers together? Does this underestimate the tobacco burden and the resultant outcomes of the study?
  6. There is a typo on line 269 – ‘unhealthy diet’

Reviewer 3 Report

1.  Reviewed article submitted contain the required sections: Author Information, Abstract, Keywords, Introduction, Materials & Methods, Results, Conclusions, Figures and Tables with Captions, Funding Information, Author Contributions, Conflict of Interest and other Ethics Statements.

2. Only 22 of the 33 references cited in the article are recent (over the last 5 years).

3. The results of the conducted research correspond to the set goal, the materials and research methods used are described in detail, statistical methods correspond to the purpose of the study.

4. The article is written in a clear language, the data presented in the tables are correct, reflect the scientific results presented in the text.

5. The frequency of simultaneous presence of two or more modifiable risk factors for cardiovascular diseases in the studied cohort of the population of Burkina Faso was analyzed.

I have some questions:

1. It did not seem quite logical to analyze the data obtained in 2013 (before the implementation of the first NCD plan drafted in 2016). As a rule, such an analysis precedes the development of population-based prevention programs.

2.  How can you explain the high prevalence of associated risk factors, especially among those with a high level of education in Burkina Faso?

Many epidemiological studies have shown that people with higher education have better access to health and social resources, better opportunities to live and work in an environmentally friendly environment, and lead healthy lifestyles. In contrast, people with lower levels of education may be less informed about risk factors and opportunities for prevention of CVDs, and may be less attentive to their health and less able to interpret and comply with medical advice.

Round 2

Reviewer 1 Report

Thanks to the authors for paying attention to my comments.

The authors tried to set the study's hypothesis, but it did not go down so well. The hypothesis is formulated backward (i.e., based on the results), so it sounds like it has no biological sense. Table 2 does not carry a scientific payload, presents a co-occurrence of specific characteristics with two or more unknown risk factors, and lacks interpretation. The same goes for Table 3, which will be very difficult to cite as it is unclear which lifestyle is associated with which risk factor. Thus, it has only superficial scientific utility. In terms of applicability in interpretation and usefulness for the scientific community, the most interesting could be Table 4. However, applicability is restricted given the lack of information on the significance level and is divided only by gender, while before that, it was mentioned about age group, level of education, marital status, profession, and residence. In addition, from this point of view, it would be necessary to conduct a linear regression with these covariates rather than present intermediate data. That is why "The authors did not clearly explain what exactly they found." The discussion overflowed with unnecessary information and might be shortened.

Reviewer 2 Report

Thank you for addressing my comments from the first review. I have no further comments.

Author Response

We would like to thank you for the review of our manuscrit.